# Activity of Povidone in Recent Biomedical Applications with Emphasis on Micro- and Nano Drug Delivery Systems

**DOI:** 10.3390/pharmaceutics13050654

**Published:** 2021-05-04

**Authors:** Ewelina Waleka, Zbigniew Stojek, Marcin Karbarz

**Affiliations:** 1Faculty of Chemistry, Biological and Chemical Research Center, University of Warsaw, 101 Żwirki i Wigury Av., PL 02-089 Warsaw, Poland; ewaleka@ch.pw.edu.pl (E.W.); stojek@chem.uw.edu.pl (Z.S.); 2Faculty of Chemistry, Warsaw University of Technology, 1 Pl. Politechniki Av., PL 00-661 Warsaw, Poland

**Keywords:** directed drug delivery systems, povidone, nanoparticles

## Abstract

Due to the unwanted toxic properties of some drugs, new efficient methods of protection of the organisms against that toxicity are required. New materials are synthesized to effectively disseminate the active substance without affecting the healthy cells. Thus far, a number of polymers have been applied to build novel drug delivery systems. One of interesting polymers for this purpose is povidone, pVP. Contrary to other polymeric materials, the synthesis of povidone nanoparticles can take place under various condition, due to good solubility of this polymer in several organic and inorganic solvents. Moreover, povidone is known as nontoxic, non-carcinogenic, and temperature-insensitive substance. Its flexible design and the presence of various functional groups allow connection with the hydrophobic and hydrophilic drugs. It is worth noting, that pVP is regarded as an ecofriendly substance. Despite wide application of pVP in medicine, it was not often selected for the production of drug carriers. This review article is focused on recent reports on the role povidone can play in micro- and nano drug delivery systems. Advantages and possible threats resulting from the use of povidone are indicated. Moreover, popular biomedical aspects are discussed.

## 1. Introduction

In recent years, the treatment of many diseases is based on conventional and simple methods of drug dosage. Simple tablets, pills, capsules, creams, liquids, aerosols, suppositories, injectables, and ointments are often used [1]. Many drugs were approved despite their toxic properties and unwanted activities. This happened because the use of them was more beneficial for the organism than the potential risk posed by them [2]. Due to this fact, after the application of a specific drug, some side effects may occur. An unwanted response of the organism is often enhanced because of the long circulation time of the drug in the bloodstream and the possibility of penetration of various organs before the drug reaches the destination. Unspecified bio-distribution and lack of controllability of the drug release are serious limitations of conventional methods of treatment. It should also be noted that the death of a patient may be a result of a side effect of the anticancer treatment [3]. Moreover, the problems associated with administering conventional drugs are uncontrolled release, insufficient efficiency of the absorption process, and small bioavailability of drugs that lead to the use of higher doses. Because of this danger, the demand for improved safety of pharmacological treatment increased in recent years. Particularly intensive work was focused on drug delivery systems (DDSs). DDSs are defined as devices or preparations that enable the controlled introduction and distribution of the drug in the body [4]. The following steps can be identified in the process of construction and administration of a DDS: (a) synthesis of the carrier, (b) immobilization of the drug inside the carrier, (c) injection of the drug-currier products into the bloodstream, and (d) release of the active substance at a particular place, time, and at a specified rate. The transport of the drug-loaded carriers directly to the destination minimizes their influence on the healthy cells and the appearance of side effects. Generally, the application of DDS may eliminate/neutralize not only the toxicity to healthy cells and the attack on other organs, but also low drug solubility and the lack of biocompatibility [5]. DDSs were most commonly used to treat cancer. For this reason, in our review, we limit ourselves to this kind of illness. Nevertheless, in the literature, reports concerning the application of DDSs in other diseases can also be found.

Recently the use of micro- and nanoparticles as potential carriers for anticancer drugs has gained in popularity. The synthesized nanoparticles are inorganic or organic, such as metallic particles, magnetic particles, quantum dots, and micelles, liposomes, dendrimers and polymers (including hydrogels) [6]. Among others, hydrogels appeared to be attractive materials for drug delivery carriers. They are made of polymers and form three-dimensional networks filled with aqueous media. The substantial water-absorption capacity of hydrogels is connected with the presence of hydrophilic groups in the monomers and crosslinkers that are used in the synthesis of the polymer chains [7,8]. The interesting ability of hydrogels is the extensive change in their volume during the phase transformation, which occurs in response to a change in the environmental factors. These factors include temperature, pH, ionic strength, and concentration of specific ions [9,10]. The very small size of nanohydrogels enables their injection, distribution in the body through the bloodstream, and penetration into the cells by endocytosis. Thus far, a number of polymers have been applied to build drug delivery systems. The natural polymers, including chitosan [11,12], gelatin [13,14], polylactic acid [15,16], polysiloxane [17,18], collagen [19,20], and alginate [21,22] appeared to be good and useful materials. In addition to these compounds, the synthetic polymers were also used. They include poly(N-isopropylacrylamide) [23,24], poly(acrylic acid) [25,26], poly(N-vinylcaprolactam) [27,28], polyethylene glycol [29,30], polyurethane [31,32] and some more.

One of the very interesting materials belonging to the hydrogel class is poly (N-vinyl pyrrolidone) (pVP); it is also called povidone and polyvidone. The biomedical properties and applications of povidone are well described in the literature. Povidone entered the list of materials approved by the U.S. Food and Drug Administration and is listed in the issues of Pharmacopoeia [33]. An apparent advantage of pVP is the presence of the hydrophobic alkyl group and the polar amide group in the pyrrolidone ring. As a result, pVP is well soluble in cold water and many organic solvents, which allows the synthesis of micro- and nanoparticles under various conditions [34]. A solution of pVP is really stable; nevertheless, mold may appear in it after some time. pVP can be sterilized with steam; this does not affect its properties [35]. In solutions, pVP exhibits a high degree of compatibility/complementarity with many inorganic salts and other chemicals, which is very useful in the process of inserting drugs into the carrier particles [36]. It is an odorless and non-flammable material [37]. Moreover, pVP exhibits hygroscopic and adhesive properties [38,39]. Due to rising ecological awareness, absolutely safe materials should be selected for the pharmaceuticals. pVP is considered to be an environmentally friendly substance [40]. Therefore, it can be effectively used in the chemical and pharmaceutical industry. From the perspective of the design of DDSs and subsequent examination of the ability of the carrier to deliver the drug to the cells, another important ability of pVP is its fluorescence [41]. pVP exhibits two excitation peaks: At 285 and 330 nm, and one emission peak, at either 400 or 408 nm [42]. This ability can be employed to study the effectiveness of the drug delivery to the cells. Finally, pVP is a nontoxic, non-carcinogenic, and temperature-insensitive substance, and is well biocompatible with living tissue [43]. Despite this, it is still not very popular as the material for micro- and nano DDSs. However, recently, several reports on employing unique abilities of pVP in the design of drug carriers have appeared. Due to this fact, in this review, in addition to presenting the main properties and biomedical applications of povidone, we focus on the roles pVP can play in drug delivery systems.

## 2. Popular Biomedical Applications of pVP

Due to its antitoxic properties, pVP may be used in medicine as the coating for potentially dangerous compounds and the reducer of their unwanted activities [44]. The coating was applied in the drug called povidone-iodine (pVP-I), a formulation usually used by surgeons as a preoperative antiseptic [45]. The antibacterial, antiviral, and antifungal activities of pVP-I contributed to the selection of this drug as the first-line preventive aid to assisting the treatment of COVID-19. At present, the ongoing clinical trials demonstrate the benefits of PVP-I in intranasal and gargling use against COVID-19 [46]. In addition, pVP is a component of eye drops, epidermal, and vaginal preparations, etc. [47,48,49]. Taking into account good solubility of pVP in different solvents, it can be combined with an active substance to improve its absorbability [50,51]. pVP’s ability of swelling or its hygroscopic properties made possible the application of povidone as tablet disintegrant [52,53]. Contrary, considering the adhesive ability of pVP, it appeared to be useful as the binder for tablets [33,54]. Moreover, it was used as an ingredient of several lubricants [55]. Since pVP can absorb water, it also plays the role of a countermeasure against diarrhea in medicine [56]. As pVP has the ability of easily forming a film, it is used in contact lenses [57,58]. It has also been used as blood plasma substitute, but finally was replaced by another substance [59,60]. pVP has several synonyms. In Table 1, some clinically approved drugs and preparations with pVP as ingredient are shown. In micro- and nanoscale, povidone may be potentially useful in drug delivery systems; this will be addressed in the next section.

## 3. pVP in Micro- and Nano Drug Delivery Systems

Due to the diversity of properties of povidone, it can play several roles in micro and nano drug delivery systems. Povidone functions can be analyzed from two different perspectives. First, pVP can serve in drug delivery systems as a modifying component in the copolymer, base material, and the shell (see Figure 1). On the other hand, the tasks of povidone can be identified by analyzing the prism of activity it evokes. All major pVP activities are presented in Figure 2 and Figure 3. They include: Unwanted-toxicity reduction, stabilization of solutions, enhancement of drug bioavailability, improvement in film formation, solubilization of sparingly soluble substances, lyophilization, and generally drug delivery [61,62].

However, the building role is usually related to the activity function. Moreover, pVP can be added to the synthesis mixture due to its ability to limit the particle size and to control the particle shape during the polymerization. In Table 2 the correlation between pVP functions and potential biomedical application of drugs containing pVP in DDSs is shown.

### 3.1. pVP Coating

One type of drug nanocarriers is core-shell nanoparticles. Metal-, magnetic-, and polymeric materials are mainly used as cores. The core is primarily a magazine for the active substance. The coating/shell is usually made from polymers, including pVP. Still, the shell can fulfill various functions [63]. In the next paragraphs, the application of pVP coating was clearly described.

#### 3.1.1. pVP as Stabilizer and Size Limiter

The stabilizing properties of pVP can be seen in three different perspectives. The first one is the prevention of aggregation. For example, Foldbjerg et al. prepared the Ag-NPs solution by suspending Ag-NPs powder in ELGA water. pVP was used as the stabilizer. The toxicity was examined by means of the flowcytometric annexin V/propidium iodide (PI) assay. The results showed that both Ag^+^ and pVP-Ag-NPs induced necrosis and apoptosis in THP-1 cells. Moreover, during the exposure, the ROS were produced. The results proved that oxidative stress was an important indicator of cytotoxicity caused by Ag-NPs and Ag^+^ [64]. In another research, Guo et al. investigated the influence of pVP coated Ag-NPs on inhibition of the growth of acute myeloid leukemia (AML). Additionally, the experiment was repeated with Ag^+^. Compared to their previous research, they specified the cytotoxicity mechanism of Ag-NPs. The nanoparticles were prepared by a continuous flow electrochemical process in which only polyvinylpyrrolidone (pVP) and two silver rods were used as the stable agent and the electrodes, respectively. The influence of vitamin C and N-acetyl-l-cysteine (NAC) as protection against ROS was also investigated. The results showed that Ag-NPs were capable of bypassing typical barriers and then releasing silver ions that damaged the cell machinery. Moreover, the toxicity of Ag^+^ was much higher than that of pVP-coated AgNPs against THP-1 cells. In the case of NAC and vitamin C, the research showed that both compounds were capable of completely reversing the generation of ROS after the treatment with Ag-NPs. However, only NAC could protect the cells from more serious damages such as apoptosis [65]. In view of the increase of ROS level after the use of Ag-NPs, Fahrenholtz et al. investigated its influence on three ovarian cancer cell lines: SKOV3, A2780, and OVCAR3. The main choice for the treatment of ovarian cancer is cis-platin (cis-diamminedichloroplatinum(II), CDDP). The authors, besides the examinations with alone pVP-coated Ag-Nps, as the first ones inspected heft combined therapy with the use of prepared nanoparticles and CDDP. The results showed that Ag-Nps were highly cytotoxic against A2780 and SKOV3 cells, whereas OVCAR3 cells were less sensitive to Ag-NPs. The level of ROS was higher in A2780 and SKOV3 compared to OVCAR3. Due to that, it was possible that DNA damage occurred in A2780 and SKOV3. Moreover, the combination of CDDC with pVP- coated Ag-Nps, had the ability to reduce the number of cytostatic doses. However, the authors emphasized that the in vivo examination of the combined therapy should be done in the future [66]. The literature reports indicated that Ag-NPs can be used in tumor therapies and sensing applications and became an encouraging tool for in-vivo imaging. However, after the injection to an organism, they may be coated by protein corona pretty fast, which might lead to the deactivation of the nanoparticle before the start of its therapeutic action [67]. Due to this fact, Batista et al. used pVP as one of the coating ingredients to limit protein aggregation. The authors investigated protein absorption on the Ag-NPs surface stabilized with a mixture of pVP, poly(ethylene oxide)—PEO, and PEO-b-P2VP (poly(ethylene oxide)-b-poly(2-vinyl pyridine). The results were satisfactory [68]. pVP was also used to coat other metal particles, such as gold [69,70].

Regarding the second property, in addition to their stabilizing function, pVP shell can be useful as a size limiter [71,72]. In particular, in paper [73], Au-NPs were synthesized via the addition of Cetyl trimethylammonium bromide (CTAB) solution to HAuCl_4_. Next, Au-NPs were functionalized through the dynamic stirring method with pVP. Both Au-Nps and pVP coated Au-NPs were characterized using HRTEM and zeta potential. The result showed that zeta potential increased from −34.3 mV for free Au-Nps to −18.7 mV for pVP-coated nanoparticle. HRTEM analysis showed a slight increase in the size of the nanoparticles from 13 to 14.2 nm that was associated with the presence of the polymer. Despite the magnification of nanoparticles, at the same time, the reduction of the size of the metallic core was observed. The results confirmed the size-limiting ability of pVP. The obtained nanoparticles were connected with doxorubicin (Dox@pVP-AuNPs) and used in the treatment of human lung cancer. It appeared that Dox@PVP-Au-NPs were more cytotoxic compared to free DOX. It was caused by the highly effective cellular entry of Dox@PVP-Au-NP. Moreover, Dox@ PVP-AuNPs increased ROS generation, induced both early and late apoptosis in the lung cancer cells, and sensitized mitochondrial membrane potential.

#### 3.1.2. pVP as Shape Controller

Apart from the anti-aggregation and size-limiting ability of pVP, its stabilizing properties involve shape control of the synthesized nanoparticles [74,75]; this is the third property of pVP. For this reason, Yang et al. [76] developed a facile approach to the synthesis of ZnO in different shapes via the chemical precipitation method in water. Contrary to the previous reports, their method was eco-friendly and eliminated the use of organic solvents. For the synthesis, a mixture of zinc nitrate, urea, and pVP was used. The achievement of three different shapes, such as flower, bowtie, and nest, was possible thanks to the use of different concentrations of pVP. With the smallest concentration of pVP, the bowtie-shaped particles were formed. The process involved three distinct phases. Firstly, star-like particles were formed. Next, the formation of densely packed sharp branches on each side of the ZnO-particles was observed. Finally, the branches merged into flakes, leading to bowtie-shaped particles. In the case of using an intermediate concentration of pVP the particles formed the flower-like shape. At first, the porous spheres were produced. Then, the obtained particle served as a template for the growth of petals. Finally, using the greatest concentration of pVP, the nest particles were observed. The obtained nanoparticles were toxic against *E. coli* and *S. aureus*. The antibacterial activity depended not only on the morphology of the nanoparticles but also on their surface area, pore size and the used components. However, the killing-bacteria ability appeared mainly due to the release of Zn^2+^ and the generation of corresponding ROS-independent oxidative stress.

#### 3.1.3. pVP as Cryoprotective Agent

The interesting property of pVP was noticed by Linde Laboratories. They proposed to use this polymer as a substance able to protect cells from freezing injury. This can be possible through plugging the pores in the cells with their coating. This action prevented the cells from overdehydration, ice formation, and changes in the intracellular electrolyte concentration. In addition, pVP allowed lyophilization of biological materials when glycerol was used [77]. This research was groundbreaking; pVP’s ability was used in many clinical applications, among others, in the lyophilization of cells. The cryoprotection abilities were also used in micro and nano drug-delivery systems. Dorati et al. studied the lyophilization process of nanoparticles based on polylactide-co-glycolide (PLGA-H) and polylactide-co-glycolide-co-polyethylenglycol (PLGA-PEG) in a ratio of 30:70. The nanoparticles were obtained via the solid-oil-in-water technique. As cryoprotection agents, pVP-K17, pVP-K32 and sodium carboxymetylcellulose were used. The results showed that the use of polymers stabilized the nanoparticles during the freeze-drying process. In consequence, the aggregation did not occur. All formulations containing cryoprotectants exhibited desired properties after their lyophilization, such as correct size and shape. In addition, in the samples with pVP-K17 and pVP-K32, the collapse phenomenon was not present. The authors suggested that the best protection of the particles against freeze destruction was a blend with pVP-K17 and pVP-K32. In another work, before the lyophilization process, the nanoparticles were loaded with gentamycin—an antibiotic used to treat several types of bacterial infections. One of the important aspects concerning the stability of nanoparticles is zeta potential. The solution of gentamicin sulfate loaded NPs is characterized by zeta potential close to the neutral value (from 0.5 to 3.58 mV), which corresponds to unstable suspensions. After lyophilization, the redispersed gentamycin sulfate-loaded NPs had a slightly negative zeta potential, which was caused by cryoprotectant interactions. However, the neutral zeta potential has a positive impact on antimicrobial activity [78]. Mir et al. examined the influence of pVP concentration (0.5 ÷ 2% *w*/*v*) on the size of the carvacrol-poly(caprolactone) nanoparticles after the lyophilization process. The results indicated that with a high pVP concentration (1 ÷ 2%), the size of the nanoparticles decreased; however, the NPs aggregated. For a small pVP concentration (0.5%), the size of NPs was the same as before the lyophilization, and the NPs were freely flowing [79]. Gaber et al. compared three types of cryoprotectant agents, such as hydroxypropyl-β-cyclodextrin (HPβCD), pVP, and trehalose, in the lyophilization process of albumin nanocapsules (NC) and nanoemulsion (NE). The obtained results indicated that the presence of 5% pVP led to the formation of fluffy powders of both NC and NE. However, after the lyophilization process, the size of NE diminished, while the size of NC increased. For 2.5% concentration of pVP the fluffy powders were not observed. The products, after lyophilization, were non-dispersible and viscous [80]. The mentioned above research data confirmed the cryoprotective abilities of pVP. Finally, depending on the substances used, the concentration of pVP should be either elevated or decreased.

#### 3.1.4. pVP as Unwanted-Toxicity Reducer

The antitoxic activity of pVP can be useful for reducing the adverse influence of the drugs and other substances on the patient body. As was already mentioned, some metal-oxide nanoparticles are exploited for the delivery of medicines. However, their use led to significantly lowered cell viability. İşçi et al. checked the influence of pVP on the toxicity of iron(III) oxide nanoparticles. For this purpose, they synthesized Fe_3_O_4_ nanoparticles with pVP shells. The obtained results showed that, compared to non-covered Fe_3_O_4_, the PVP coating of the nanoparticles increased the protection of the healthy osteoblast cells. Moreover, the addition of polymeric shells did not affect the superparamagnetic character of the iron oxide [81]. One complication occurring in patients suffering from neoplastic diseases is venous thromboembolism. The main cause of venous diseases is neutrophil extracellular traps (NETs). They are also an important factor in the development of the tumor growth process. The literature reports say that some anticancer drugs, e.g., 5-fluorouracil (5FU), can escalate the problem with creating NETs. For this reason, Basyreva et al. developed protection of patients against side effects of 5FU chemotherapy. They prepared amphiphilic poly-N-vinylpyrrolidone (Amph-pVP) and used it as a 5-fluorouracil (5FU) carrier. The obtained data showed that 5FU alone increased two to three times the total amount of NETs in the blood. However, when Amph-pVP was used, the adverse effect of 5FU was completely blocked, and the formation of NETs was not observed [82]. The pVP coating can play several roles. For example, Chen et al. prepared the bortezomib (BTZ) nanoparticles with tannic acid covered by pVP shell. Firstly, it seemed that polymer only stabilized nanoparticles. The results showed that the addition of pVP made them stable over the entire time of the experiment (15 days). This system also appeared to be useful in antitumor treatment. The cytotoxicity examinations indicated that the modified BTZ nanoparticles were more cytotoxic than BTZ alone. Moreover, the analysis of the distribution of the drug in model tumor-bearing mouse led to a conclusion that a higher concentration of the drug in the tumor tissue was found when BTZ was covered with pVP. Correspondingly, the modified BTZ nanoparticles caused superior therapeutic effects. Next, in vitro examination indicated pVP as a substance for improving the bioavailability of BTZ. However, in vivo experiments showed that mice treated with BTZ alone were characterized by weight loss while the pVP coated BTZ-NPs led to weight gain in mice. This phenomenon indicated the reduced systemic toxicity of the BTZ-NPs [83].

#### 3.1.5. pVP for Protection and Solubilization of Substances and for Enhancing Solubility

The pVP polymer can protect healthy tissues from toxic substances, improve the solubility of the drug and enhance its bioavailability. For example, variety of structures/compositions of pVP is responsible for its solubility in many different substances. This property can be used to increase the bioavailability of some drugs after loading them into the polymer net. The drug solubility in aqueous solutions is enhanced through the formation of the drug-polymer complex [84]. This feature can be employed in building specific drug delivery systems acting against, e.g., neoplastic cells. In this respect, Tiwari et al., in their research, used graphene nanoparticles coated with pVP as quercetin (QSR)- and gefitinib (GEF) drug carriers. Moreover, the influence of the combination of these drugs on ovarian cancer cells was investigated. GO-NPs were connected with pVP in a carbodiimide-activated esterification reaction. The results showed that the functionalization of GO-NPs surface with the selected polymer, pVP, increased drug solubility, and biocompatibility. In addition, the combined drugs improved cytotoxicity against the ovarian cell [85]. On the other hand, it was shown that the addition of pVP to GO did not influence the cytotoxicity of graphene oxide significantly, contrary to the popular belief that pVP acts as the toxicity reducer [86]. In another paper, Ozkan et al. prepared pVP particles connected with the drug through the supercritical antisolvent process (SAS). Quercetin and rutin were used as the model substances. These compounds belong to the class of flavonols and are characterized by anti-inflammatory, antioxidant, and antiviral activities. The main disadvantage of flavonols is their poor solubility in water, which results in their weak bioavailability. The obtained results proved that the use of the polymer led to an increase in the dissolution rates by circa 10 times for quercetin and 3.19 times for rutin, compared to flavonols alone. In addition, high effectiveness of connection of the drug with the polymer was achieved [87]. Homayouni, using pVP-K30 and HPMC (hydroxypropyl methylcellulose), obtained curcumin (Cur) nanoparticles by antisolvent crystallization. Cur is anticancer, antioxidant, anti-inflammatory, and antimicrobial. However, it just, as flavonols, is characterized by poor water solubility, which in consequence limits its oral bioavailability. It appeared that by using pVP the solubility of curcumin was increased. The time of dissolution was shortened from circa 20 min for curcumin alone to circa 7 min for the nanoparticles with pVP. Moreover, the examinations revealed that pVP inhibited the aggregation of the particles and that the connection between Cur and pVP is realized by hydrogen bonding [88]. The literature papers report that anticancer curcumin acts like inhibitor for STAT3 and NF-κB signaling pathways, which plays the main role in cancer development. In addition, Cur can counteract the activation of Sp-1 and its downstream genes such as EPHB2, ADEM10, HDAC4, calmodulin, and SEPP1 [89]. Moreover, Cur solubility can be improved when one of its chains is connected with a metallic nanoparticle while another chain is attached to one of the side chains of such soluble polymers as inter ilia pVP. For this purpose, the folate–curcumin-loaded gold–polyvinylpyrrolidone nanoparticles (FA–CurAu-PVP NPs) were prepared. The drug and the targeting substance were connected with Au-NPs using the layer by layer assembling. FA–CurAu-PVP NPs were characterized with UV–Vis spectra, FT IR spectroscopy, X-ray powder diffraction, and thermogravimetric analysis. In vitro analysis was done using the MTT assay. Additionally, the in vivo test of NPs against mouse-model breast cancer was examined. The results suggested that the obtained nanoparticles seemed to be a promising candidate for Cur chemotherapy to destroy the tumor cells without harming normal cells [90].

From the drug-delivery perspective, one of the most important properties of polymeric carriers is their biocompatibility with the tissue. Thus far, a rather big number of reports on compliance of pVP have appeared. However, there are only a few research papers on the performance of this polymeric carrier in the blood. Tsatsakis et al. checked the behavior of Amph-pVP nanoparticles in the presence of whole blood, blood plasma, and blood cells. Moreover, they evaluated the effects of these nanoparticles on endothelium cell growth/viability. Amph-pVP was obtained via self-assembling in an aqueous medium. pVP formed a hydrophilic shell placed over the hydrophobic alkyl core. It was found that Amph–pVP had no effect on the coagulation process, complement activation, and platelet aggregation in the tested concentration range of 0.05 ÷ 0.5 mg/mL. In addition, no considerable hemolytic or inflammatory effect was observed [91]. Pornpitchanarong et al. prepared two types of nanoparticles based on pVP and acrylic acid: Copolymer and core-shell structure. The surfactant-free emulsion polymerization was used. The size of nanoparticles was in the range 138 ÷ 183 nm. The obtained nanoparticles were loaded with cis-platin via a coordinate covalent bond, and relatively high drug loading was achieved. The cytotoxic experiments showed higher toxicity of cis-platin loaded nanoparticles against the cancer cells compared to the drug alone. Since the nanoparticles guaranteed sustained, moderate release of the drug, they can be used in long-term therapy, which is safer for the patients [92]. Xiang et el. developed the reduction method to synthesize AuPd@pVP NPs in the presence of pVP. The polymer was used as both surfactant and stabilizing agent. They reported that thanks to the formation of pVP shell, good biocompatibility both in vivo and in vitro were possible, and the obtained nanoparticles were useful in photothermal therapy (PTT) under NIR laser irradiation of low power [93]. Mahdavi et al. prepared gadolinium oxide nanoparticles covered with pVP. This radiosensitized core-shell system could be combined with a cytostatic drug. In consequence, it was possible to use this multifunctional drug delivery system in tumor treatment. The authors confirmed that the surface modification of Gd_2_O_3_ NPs with PVP could increase their biocompatibility. The obtained results showed that the viability and surviving fraction of B16F10 melanoma cells were significantly inhibited when Gd_2_O_3_ @ PVP-DOX NPs under radiation were used compared with treating with the free DOX or PVP-Gd_2_O_3_ NPs alone under radiation [94].

Thenmozhi concluded that the functionalized iron oxide nanoparticles induced apoptosis in MCF-7 breast cancer cell lines. Iron oxide nanoparticles were coated with PVP by adding PVP to a suspension of nanoparticles, sonicating the solution for 30 min, and thermal annealing the separated product at 600 °C for 30 min. Finally, the PVP coated iron nanoparticles were biofunctionalized with Syzygium aromaticum flower bud’s (clove) extract. The author found that a significant quantity of the nanoparticles loaded with S. aromaticum extract was in the cells during the sub-G1 state. In addition, the nanoparticles influenced the caspase–3, caspase–8 and caspase–9 responses. The oxidative stress caused by ROS formed after the treatment with nanoparticles was also observed. The pVP coating played not only the role of the stabilizer of iron oxide nanoparticles but also improved the bioavailability and therapeutic effect of plant extract [95].

### 3.2. pVP in Copolymers

#### 3.2.1. pVP as LCST Stabilizer

pVP often worked as a copolymer. From the point of view of drug delivery, in the case of temperature-sensitive systems, the LCST (lower critical solution temperature) close to the physiological level (37 °C) should be achieved. Effective change of LCST can be obtained after adding a hydrophilic monomer to the reaction mixture. For example, a copolymer poly(N-isopropylacrylamide-co-N-vinylpyrrolidone) was synthesized. The best results were obtained for a monomer molar ratio (NIPAAm/NVP) of 91.5:8.5 in the copolymer. Next, the polymerization suspension was used to form the microspheres. Finally, the microspheres were loaded with a model drug—diclofenac. The release of that drug through the pulsatile mechanism was investigated [96]. In another try, a new microgel based on acrylamide (AAM), 1-vinyl-2-pyrrolidone (NVP), and 2-(diethylamino)ethyl methacrylate (DEAEMA) was prepared. The free-radical precipitation polymerization was applied. The obtained microparticles increased the LCST temperature from 32 °C to circa 38 °C (physiological level). 5-FU was loaded into the microgel with efficiency of 32.47 mg/g. The obtained results indicated that 5-Fu was released slightly faster at pH 7.4, and the microgel did not exhibit visible cytotoxicity [97].

#### 3.2.2. pVP as Solubilizing Agent

An interesting application of pVP copolymer was proposed by Swilem et al. They synthesized nanogels based on poly(acrylic acid)-co-N-vinylpyrrolidone using the ionizing radiation method. The obtained material was used as a tear substitute; it could be useful in treatment of dry-eye syndrome. Poly(acrylic acid) is a major component of various brands of artificial tears and is characterized by mucoadhesive property and high water absorbance. However, there were disadvantages of using this kind of polymer; viscous films could be formed and they caused transient blurring of vision and sticky sensations. The addition of pVP reduced viscosity of tears significantly and, therefore, increased the comfort of the patient. Moreover, the use of the copolymer enhanced markedly the effectiveness of dry eye curing [98].

#### 3.2.3. pVP as Base Material

pVP can also be useful as the base material. As already mentioned, pVP is a hydrophilic polymer, which can improve biocompatibility and solubility of many drugs. However, only several reports on pVP as the main ingredient of the nanoparticles appeared in the literature. For example, Peng et al. synthesized, with the addition of N-vinylformamide to the main monomer, the pVP material using the reversible addition–fragmentation chain-transfer (RAFT) polymerization. The obtained material was sensitive to redox actions and degraded in the presence of GSH. The nanogel could be used to carry doxorubicin (DOX). A decent level of drug loading was achieved. The in vitro examinations indicated that blank nanogels were biocompatible, and DOX-loaded nanoparticles exhibited medium antitumor activity [99]. Prosapio et al. proposed a method of obtaining drug/polymer microparticles, which was based on coprecipitation of the active substance with a hydrophilic polymer (pVP) using a supercritical antisolvent. Nimesulide was used as a model drug. The size of microparticles was in a range from 1.7 to 4 μm. The drug-release examination indicated that by using pVP the dissolution rate could be enhanced by circa 2.5 times [100].

## 4. Limitations of pVP

There have been only several reports on rare cases where pVP caused side effects in the patients. It acted as an allergen and caused urticaria, dyspnoea, and anaphylaxis [101,102]. For this reason, the particles of high molecular weight, multiple doses, and injections of large amounts of pVP in the same place should be avoided. Regarding the use of pVP during intracytoplasmic sperm injection as the improver of viscosity of the sperm solution, pVP may cause changes in the sperm structure and its visibility in the injected embryos during the early development period [103]. pVP, in general, is classified as a biodegradable material. However, there are several reports in the literature on the degradation problem and in consequence the limitation for parenteral administration. Because of that, low molecular weight pVP should be useful for parental administration, due to its possibility of being excreted through the kidneys [104].

## 5. Conclusions and Future Perspectives

Despite many advantages and a wide use in medicine, compared to other polymers, pVP is not a very popular material in drug delivery systems. Thus far, pVP is mainly used in drug delivery systems as the coating or shell ingredient (see Table 2), the protection from aggregation and the shape and size controller. Few studies were concerned with nano- and microhydrogels, including pVP as a copolymer and its ability to change LCST of the main polymer. However, due to an increase in demand for new biomaterials and the use of micro- and nano drug delivery systems, the situation should change soon. In addition, pVP has other abilities, which can be used in the production of new copolymers with new properties. Thus far, just a few reports on the application of pVP as the main component in the carriers have appeared. A pending challenge is to use pVP in micro- and nanomedical devices as the main building material. The literature reports are focused mainly on the nanoparticle synthesis employing the electron beam method; this method requires rather expensive, specific equipment. Moreover, there is some danger related to the emitted X-ray radiation. In the future, development of simpler and more efficient methods of synthesis of the pVP particles may lead to a significant progress in the construction of pVP micro- and nano drug-carriers.

## Figures and Tables

**Figure 1 pharmaceutics-13-00654-f001:**
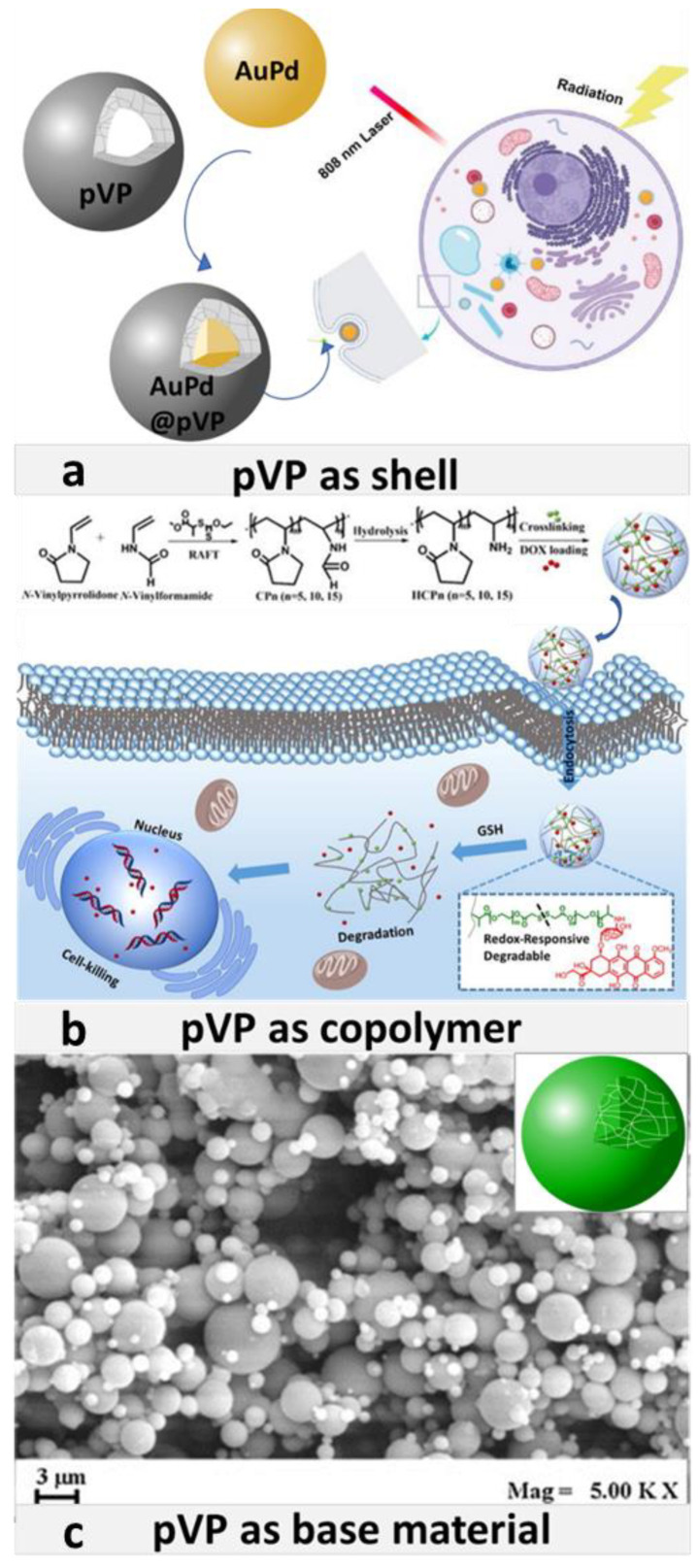
(**a**) Scheme of the AuPd core and pVP shell in AuPd@pVP nanoparticles, and their application in photo-thermal therapy. Adapted with permission from [93], Elsevier, 2020. (**b**) Scheme of synthesis of pVP and N-vinylformamide copolymer and way of its operation against cancer cells. Adapted with permission from [99], Elsevier, 2019. (**c**) Field Emission Scanning Electron Microscope image of pVP microparticles precipitated from DMSO at 90 bar, 40 °C, 30 mg/mL and obtained using SAS method. Adapted with permission [100], Elsevier, 2016.

**Figure 2 pharmaceutics-13-00654-f002:**
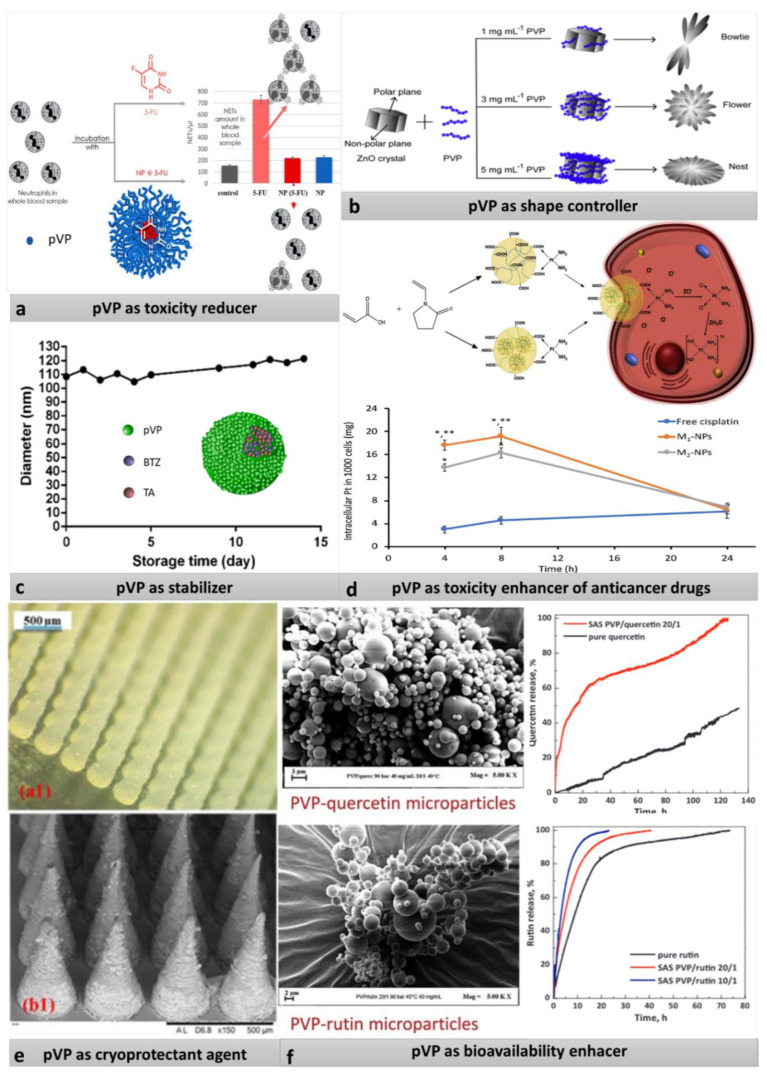
(**a**) Scheme of reduction of 5-fluorouracil of neutrophil extracellular traps by closing the drug into Amph-pVP. Adapted with permission from [82], Elsevier, 2020. (**b**) Influence of different concentration of pVP on the shape of ZnO-PVP nanoparticles. Adapted with permission from [76], Elsevier, 2019. (**c**) Antidegradation action of pVP. Adapted with permission from [83], Springer, 2020. (**d**) Scheme of the action of nanoparticles obtained by two different methods, and graph showing enhancement of cisplatin toxicity through increased intercellular accumulation of drug bonded with NP compared to free cisplatin (*—statistically significant difference (*p* < 0.05) from free cisplatin, **—statistically significant difference from M2-NPs). Reproduced with permission from [92], Elsevier, 2020. (**e**) Digital microscope (a1) and SEM (b1) images showing formulation of microneedles prepared from carvacrol- poly(caprolactone) nanoparticles with pVP as cryoprotectant agent. Reproduced with permission from [79], Elsevier, 2020. (**f**) Increased bioavailability of quercitin and rutin via covering drug with pVP. Reproduced with permission from [87], Elsevier, 2019.

**Figure 3 pharmaceutics-13-00654-f003:**
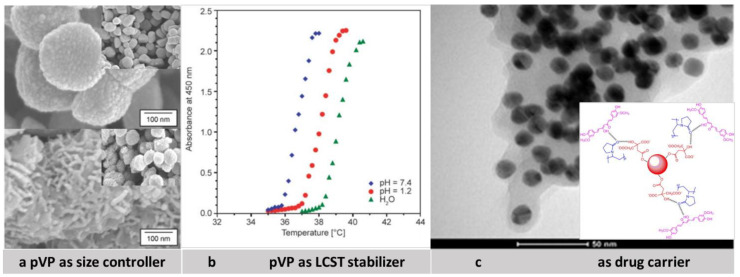
(**a**) Scanning electron micrographs of iron oxide powders synthesized in ethylene glycol at 140 °C for 2 h without pVP (top picture) and with pVP (bottom picture). Adapted with permission from [72], Elsevier, 2018. (**b**) LCST profiles of poly(NIPAAm-*co*-NVP) nanoparticles under simulated physiological conditions showing influence of pVP on LCST of polymers. Reproduced with permission from [96], Budapest University of Technology and Economics, 2019 (**c**) TEM images of pVP functionalized gold nanoparticles with curcumin and scheme of possible conjugation of Cur with pVP coated Au-Nps. Reproduced with permission from [62], Elsevier, 2012.

**Table 1 pharmaceutics-13-00654-t001:** List of selected clinically approved drugs containing pVP.

No.	Name of Drug	pVP Type	Role of pVP	Therapeutic Effect of Drug
1	FRESHKOTE^®^ Preservative Free	povidone K30	lubricant	delivers relief from dry eye symptoms
2	BD PurPrep™	povidone-iodine	antiseptic	patient preoperative skin preparation
3	Braunol 2000	povidone-iodine	antiseptic	wound antiseptic
4	Betadine^®^ VAG	povidone-iodine	germicidal	Antiseptic in acute and chronic vaginitis
5	FUCIDIN^®^	Crospovidone and povidone 90f	super-disintegrant and binder	antibiotic with antimicrobial activity
6	Neozipine XL	povidone K25 and crospovidone	bioavailability improver and super-disintegrant	used to treat high blood pressure
7	Cefdinir	povidone K30	binder	used to treat a wide variety of bacterial infections
8	OXYCONTIN^®^	povidone K30	binder, solublizer	used to mitigate persistent pain
9	Pantoprazole Sodium Delayed Release	povidone K25 and crospovidone	Solubilizer and used to allow absorption of the active drug.	used to treat gastroesophageal reflux disease
10	Tylenol (chewables)	crospovidone	taste masking	painkiller for children

**Table 2 pharmaceutics-13-00654-t002:** Examples of application of pVP as nanocarriers.

Functions of pVP		Name of Drug Carrier	Name of Drug	Therapeutic Effect of Drug	Reference
Action Functions	Bulding Functions				
drug carrier	base material	pVP microspheres	nimesulide	analgesic, anti-inflammatory, antipyretic effect	[100]
LCST stabilizer	copolymer	p(AAM-co-NVP-co-DEAEMA) microgel	5-FU	anticancer	[97]
solubilizing agent	copolymer	pVP/AAc nanogels	-	delivers relief from dry eye symptoms	[98]
solublizing agent	shell	pVP GO-NPs	QSR	anticancer	[86]
toxicity enhancer	shell	pVP/AAc core-shell nanogels	cis-platin	anticancer	[92]
toxicity reducer (prevention from side effects of drugs)	shell	Amph-pVP	5-FU	anticancer	[82]
bioavailability enhancer	shell	pVP coated Fe3O4-NPs	S. aromaticum extract	anticancer	[95]
stabilizer: Prevention from aggregation	shell	pVP coated BTZ/TA nanoparticles	BTZ	anticancer	[83]
stabilizer: Shape control	-	ZnO nanoparticles	-	-	[76]
stabilizer: Size limiter	shell	pVP coated Au-NPs	DOX	anticancer	[73]
cryoprotectant agent		PEG-PLGA/PLGA-H Nanoparticles	gentamycin sulfate	anticancer	[78]

## Data Availability

Not applicable.

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
