# Peer review of "Activity of Povidone in Recent Biomedical Applications with Emphasis on Micro- and Nano Drug Delivery Systems"

_pharmaceutics, 2021, doi:10.3390/pharmaceutics13050654_

Round 1
Reviewer 1 Report
Review Report
R43 – can are inorganic or organic
R132 – materials instead of nanoparticles
R134 – reference must be indicated
R136-142 – the influence of PVP as a stabilizer is not discussed
R173 - reference must be indicated
R189 – the paragraph needs more references from the literature
R213 – what are the organic objects
R222 – what kind of good properties
R225 – what reveals the loading with gentamicin?
R249-250 – the conclusion is not bound to the information presented above
R276 – for enhancing the solubility
R284 – what are the selected polymers?
R304 – acts like what?
R344-352 – more information about the role of PVP is needed
The manuscript must focus on the correlation between the the research data and what the scientific activity can gain for future research. It must emphasize the role of PVP and not the role of the drug delivery systems. The manuscript must be improved.
Reviewer 2 Report
The article “Activity of povidone in recent biomedical applications with emphasis on micro- and nano drug delivery systems” is focused on the efficacy of povidone in drug delivery systems.
However, the following issues must be addressed:
- Re-write the abstract by stating the problem, the limitation associated with other polymers used for drug delivery systems, and the advantages of povidone that make it useful in the design of drug delivery systems.
- The construction of sentences should be improved.
- The introduction was not well introduced. The authors should have stated the problems associated with administering conventional drugs.
- The authors stated that “It should also be noted that the death of a patient may be a result of either the disease itself or a side effect of the anticancer treatment [2].” Which is not appropriate where it was stated because a general sentence was made.
- The advantages of DDS should also be included in the introduction.
- The biomedical properties of povidone, making it useful for the preparation of DDS should be clearly stated in the introduction section.
- A table on drugs clinically approved/used and prepared with pVP should be included in the manuscript. The therapeutic outcomes of the drugs should be clearly stated in the table.
- Therefore, in the ms., the action functions were fitted to each building function?????
- There is a need for a section on future perspectives.
- The correct referencing style required by “Pharmaceutics” should be used for the references list.
- More information on pVP coating is required.
- Guo et al should be corrected. The letter “o” should be lower case.
- Tables that give a summary of pVP as stabilizer and size limiter, as shape controller, as cryoprotective agent, as unwanted-toxicity reducer, for protection and solubilization of substances and enhanced bioavailability, LCST stabilizer, as solubilizing agent, as base materials should be added to the manuscript.
- The limitations of pVP should be clearly stated, and how to overcome the limitations should also be stated in the manuscript.
- More figures are needed for illustrations.
Round 2
Reviewer 1 Report
All my remarks were taken into consideration.